# Contrast Agents for Photoacoustic Imaging: A Review Focusing on the Wavelength Range

**DOI:** 10.3390/bios12080594

**Published:** 2022-08-03

**Authors:** Seongyi Han, Dakyeon Lee, Sungjee Kim, Hyung-Hoi Kim, Sanghwa Jeong, Jeesu Kim

**Affiliations:** 1Departments of Cogno-Mechatronics Engineering and Optics & Mechatronics Engineering, Pusan National University, Busan 46241, Korea; hsi1748@pusan.ac.kr; 2School of Biomedical Convergence Engineering, Pusan National University, Yangsan 50612, Korea; dakyeon@postech.ac.kr; 3Department of Chemistry, Pohang University of Science and Technology (POSTECH), Pohang 37673, Korea; sungjee@postech.ac.kr; 4Department of Laboratory Medicine and Biomedical Research Institute, Pusan National University Hospital, Pusan National University School of Medicine, Busan 49241, Korea

**Keywords:** photoacoustic imaging, contrast agent, wavelength, spectroscopic analysis, contrast-enhanced imaging

## Abstract

Photoacoustic imaging using endogenous chromophores as a contrast has been widely applied in biomedical studies owing to its functional imaging capability at the molecular level. Various exogenous contrast agents have also been investigated for use in contrast-enhanced imaging and functional analyses. This review focuses on contrast agents, particularly in the wavelength range, for use in photoacoustic imaging. The basic principles of photoacoustic imaging regarding light absorption and acoustic release are introduced, and the optical characteristics of tissues are summarized according to the wavelength region. Various types of contrast agents, including organic dyes, semiconducting polymeric nanoparticles, gold nanoparticles, and other inorganic nanoparticles, are explored in terms of their light absorption range in the near-infrared region. An overview of the contrast-enhancing capacity and other functional characteristics of each agent is provided to help researchers gain insights into the development of contrast agents in photoacoustic imaging.

## 1. Introduction

The visualization of physiological responses in living tissues is one of the key diagnostic methods in biomedical studies [1]. Biomedical imaging techniques, such as magnetic resonance imaging (MRI) [2], positron emission tomography (PET) [3], X-ray computed tomography (CT) [4,5,6], ultrasound (US) imaging [7,8,9], and optical imaging [10,11,12] have been widely used in preclinical small-animal studies. Among these modalities, optical imaging techniques offer several advantages, including strong optical contrast, the absence of ionizing radiation, cost-efficient system configuration, and the capability of real-time imaging. Optical imaging can also provide functional information at the molecular level if multispectral data is acquired using multiple wavelengths of light [13]. However, the optical imaging techniques have limited applicability to shallow regions (~1 mm) owing to the strong light scattering in biological tissues.

Photoacoustic imaging (PAI) is a widely used biomedical imaging technique that compensates for the shallow penetration depth of pure optical imaging methods [14,15,16]. In PAI, signals are generated by optical absorption and delivered as acoustic (i.e., ultrasound) waves. Therefore, PAI inherits advantages from both optical and US imaging techniques [17,18]. Similar to optical imaging techniques, PAI is safe, cost-efficient, and easy to implement. More importantly, PAI is also capable of functional analysis of biological tissues using multispectral data acquisition [19,20,21,22]. One unique advantage of PAI for achieving adequate images is its scalable spatial resolution and imaging depth according to the application [15]. To achieve high-resolution images (~5–50 μm) in shallow regions (~1 mm), the excitation light must be tightly focused on the specimens, resulting in images of ears [23], brains [24,25,26,27], and eyes [28,29,30] in small animals. In an alternative system, light is moderately focused, or even diffused, to increase the imaging depth (~10–20 mm) at the expense of the resolution (~100–500 μm), providing whole-body images of small animals in vivo [31,32,33,34,35,36,37,38]. Recently, PAI has also been applied in clinical human studies by combining it with clinical US machines to implement real-time imaging platforms [39,40,41,42,43,44].

In PAI, endogenous chromophores that absorb light, such as oxy-hemoglobin, deoxy-hemoglobin, melanin, and lipids, can be used as contrast agents for visualizing physiological responses [45,46,47,48]. However, endogenous chromophores are limited to optically transparent organs, including tumors, the lymphatic system, and the bladder. In addition, two types of hemoglobin typically dominate all photoacoustic (PA) signals. Therefore, contrast-enhancing techniques using a variety of exogenous agents have been extensively studied, including organic dyes, gold nanoparticles, carbon nanostructures, semiconducting nanoparticles, and fluorescent proteins [49,50,51,52].

Herein, the exogenous agents used for contrast-enhanced PAI are reviewed in terms of their absorption wavelengths. First, the principles of PAI are introduced, including the parameters for signal generation efficiency. Subsequently, contrast-enhanced PAI results are introduced considering the wavelength range of the optical absorption characteristics of the exogenous agents. This review provides researchers insights into the selection of proper contrast agents for specific biomedical applications, particularly considering the wavelength selection.

## 2. Principles of Photoacoustic Imaging

PAI is based on the PA effect, which involves optical absorption and heat release (Figure 1a) [53]. When a short (typically of the order of a few nanoseconds) pulsed laser is used to irradiate a specimen, light-absorbing molecules absorb the light energy. The absorbed light energy excites the electrons in the molecules from a stable ground state to an excited energy level. The excited electrons quickly return to their ground state by emitting energy (Figure 1b). These energy emissions can be measured using two methods. The first is fluorescence (FL) light emission, which is used to implement fluorescence imaging (FLI) and two-photon microscopy (TPM). The second method is thermal energy release, which induces thermoelastic expansion of the surrounding tissue. These two types of energy release typically occur in a mixed form, rather than as purely one or the other. The heat conversion efficiency (i.e., the ratio of heat release) can be expressed as follows:(1)$$\sigma =1-\phi =\frac{k_{h}}{k_{f}+k_{h}}$$
where σ is the heat conversion efficiency, ϕ is the FL quantum yield (i.e., the ratio of FL light emission), $k_{f}$ is the rate constant for FL emission, and $k_{h}$ is the rate constant for heat release. The released heat rapidly dissipates because of the short pulse width, producing thermal vibrations that generate acoustic waves called PA waves. PA waves propagate through biological tissues and are subsequently detected by conventional US transducers. The captured signals are reconstructed in the form of images using image processing algorithms, which are similar to conventional US image processing methods [54,55,56,57,58,59,60].

Compared to the light-in-light-out principle of FLI, which suffers from low resolution in deep tissues owing to the two-way diffusion of photons, the light-in-sound-out principle of PAI can greatly enhance the imaging depth in deep tissues. Although the FLI can provide high-resolution images in biological tissues, it is highly challenging to achieve such image quality beyond the optical mean transport path (~1 mm). Recently, there have been studies for FL signal detection in deep tissues [61], but they are still limited in image acquisition. In contrast, PAI can achieve images a few centimeters deep with a resolution of several hundred micrometers with the help of acoustic wave propagation, which is less diffusive compared to the use of photons.

The initial pressure of the PA waves is linearly proportional to four parameters and is expressed as follows [62]:(2)$$P\propto \Gamma \left(T\right)\cdot \sigma \cdot \mu_{a}\cdot F$$
where P is the initial pressure of the PA waves, $\Gamma \left(T\right)$ is the Grüneisen parameter at the local temperature (T), σ is the heat conversion efficiency described in Equation (1), $\mu_{a}$ is the optical absorption coefficient, and F is the optical fluence. The amplitudes of the PA signals can be efficiently enhanced by increasing one of these four parameters. In practical PAI, the optical fluence F and optical absorption coefficient $\mu_{a}$ are controlled to increase the PA amplitude. However, the optical fluence cannot be increased above the maximum permissible exposure (MPE) of light according to the safety standard [63]. In addition, a high optical fluence does not selectively increase the PA signals of the target tissue; thus, the contrast of the resulting images is not sufficiently increased. Therefore, contrast agents with high optical absorption characteristics have been developed for contrast-enhanced PAI in various biomedical studies, including tumor imaging [64], lymphangiography [65], and drug delivery monitoring [66]. We can note that the heat conversion efficiency is residual of the FL quantum yield, as described in Equation (1). Therefore, the lower the FL quantum yield, the higher the PA signals.

The light–tissue interactions during laser excitation in PAI are highly dependent on the wavelength of the light. During the initial stages of the development of PA imaging, an Nd:YAG pumped green laser (532 nm) was frequently used as an excitation source owing to the strong intrinsic absorption of blood in the visible region [67]. Visualization of the blood vessel network with visible light sources can provide useful information for investigating hemodynamics, but the strong PA signals of blood vessels may offset signals from other chromophores. Therefore, contrast agents for PAI that absorb laser light in the first near-infrared (NIR-I, 650–1000 nm) region have been developed, which exhibit relatively low interaction with tissue, enabling photons to penetrate deeper into the biological tissues (Figure 1c) [68,69]. Using widely available lasers in the NIR-I region, multispectral PA analyses were performed to evaluate the functionalities of the developed agents. However, oxy- and deoxyhemoglobins are major absorbers in the NIR-I region. Therefore, the light absorption of PA agents has recently been extended to the second near-infrared region (NIR-II, 1000–1700 nm). In this region, photons are significantly less scattered in biological tissues, which allows for an enhanced penetration depth [70]. PA imaging with NIR-II excitation is also advantageous owing to its higher MPE for skin [63]; thus, the resulting higher optical illumination can generate stronger PA signals.

## 3. Contrast-Enhanced Photoacoustic Imaging

### 3.1. Contrast Agents at Low NIR-I (650–800 nm)

In the initial stages of the development of contrast-enhanced PAI, organic dyes were used in various biomedical studies [71,72,73]. In particular, the United States Food and Drug Administration (FDA) approved the human administration of several organic dyes, such as methylene blue (MB), Evans blue (EB), and indocyanine green (IGC). The moderate FL quantum yields of these organic dyes are suitable for PAI (Table 1); thus, they have been used for visualizing optically transparent organs, including the lymphatic system and the bladder [74,75]. The low FL quantum yields of the two blue dyes generate strong PA signals once delivered into a biological system. Song et al., successfully visualized sentinel lymph nodes (SLNs) in rats after a forepaw injection with the MB solution [76]. Compared with the control images, the contrast of the SLN was ~2, for an excitation wavelength of 635 nm. They also conducted similar experiments using EB and achieved ~2-fold signal enhancement in SLNs in rats [77]. Jeon et al., developed microbubbles in a solution of MB (MB^2^) for PA and US dual-mode imaging [78]. They showed signal changes based on the concentration of the MB^2^ solution. In addition, strong US waves can be used to burst microbubbles. Consequently, the PA signals were significantly (~2.5-fold) increased, suggesting a trigger mechanism for the control of biomedical research. In addition to blue dyes, ICG has been widely used in contrast-enhanced PAI. Kim et al., successfully delineated the lymphatic system from the blood vessel network in rats and achieved approximately 4.3-fold signal enhancement [79]. Owing to the higher FL quantum yield of ICG compared with those of MB and EB, the FL signal could also be acquired from the SLN. Park et al., acquired contrast-enhanced, dual-modal PA and FL images of bladders in rats after administration of an ICG solution [80].

In addition to contrast enhancement, different functionalities for measuring biological responses have been studied using organic nanostructures. Miao et al., developed a pH-sensitive nanoprobe using a semiconducting oligomer (SO) [84]. By synthesizing a boron-dipyrromethene dye with SO, the proposed nanoprobes were activated by the pH values of the biological tissues. The synthesized SO nanoprobes exhibited distinguishable PA responses, depending on the pH of the surrounding environment (Figure 2a). They delivered SO nanoprobes into HeLa xenograft tumors in mice and measured the PA signal differences with excitation wavelengths of 680 and 750 nm. The resulting images showed tumor-specific signals because the pH of the tumor was lower than that of the normal tissue surrounding it (Figure 2b). As a representative example of an activable probe, Reinhardt et al., developed a series of nanoprobes for the detection of nitric oxide (NO) [85]. The optical absorption peak shifted from 770 nm to 680 nm in the presence of NO (Figure 2c,d). The activation of the nanoprobe was verified by measuring the multispectral PA responses before and after injection of the developed nanoprobes into the lipopolysaccharide (LPS)-induced inflammation mouse model. The results showed significant signal enhancement compared with the control group. By visualizing NO levels in biological tissues, the role of NO in tumor progression can be elucidated in the future.

Semiconducting polymer nanoparticles (SPNs) have been developed as PA agents owing to their superior light absorption and photostability. In particular, organic semiconducting perylene diimide (PDI) was used because of its good chemical, thermal, and optical stability, as well as its good biocompatibility. Cui et al., developed PDI-based nanoparticles that could detect early thrombi [86]. They added cyclic Arg-Gly-Asp (cRGD) peptides, which have a high binding ability to glycoprotein IIb/IIIa generated from the early thrombus (Figure 2e). In the in vivo verification of the nanoparticles, the PA signals increased ~4.3-fold 24 h after injection at the early thrombosis sites in mice (Figure 2f). Yang et al., developed PDI-based nanoparticles that can act as theranostic platforms by generating reactive oxygen species (ROS) [87]. The synthesized nanoparticles release cisplatin in a tumor environment to activate nicotinamide adenine dinucleotide phosphate oxidase and trigger the conversion of oxygen to superoxide radicals, causing the generation of toxic hydroxyl radicals (Figure 2g). ROS generation by the proposed nanoparticle can be monitored using PAI owing to the strong light absorption of the ROS-sensitive molecules in nanoparticles. The resulting multispectral PA analysis showed an increase in the ratiometric PA signals of the U87MG xenograft tumor regions of mice in vivo. The PA signals at 680 nm were ~3.41-fold higher than the PA signals at 790 nm 24 h after injection, indicating the generation of ROS (Figure 2h).

In addition to organic structures, inorganic nanoparticles have been synthesized owing to the possibility of easy surface modifications for drug loading or disease targeting. In particular, gold nanoparticles (AuNPs) have been extensively studied because of their exceptionally high optical absorption efficiency [88]. Jokerst et al., synthesized gold nanorods (AuNRs), with an optical absorption peak at 760 nm, for detecting ovarian cancer [89]. They achieved contrast-enhanced images of MDA-435S xenograft tumor regions of mice in vivo. To obtain strong PA signals, Liu et al., developed AuNPs with chain vesicles by combining them with a block copolymer (BCP) [90]. Owing to the strong optical absorption of the chain vesicles, the proposed AuNPs exhibited high PA signals at 780 nm (Figure 3a). Compared with AuNPs without chain vesicles, the proposed AuNPs yielded an ~8-fold signal enhancement after subcutaneous injection into mice. Activable AuNPs were also investigated via surface modification. For example, Kim et al., demonstrated that gold/silver hybrid nanoparticles react with bacterial infections [91]. Hybrid nanoparticles were coated with a silver layer that blocked the PA signals from the internal AuNRs (Figure 3b). When the ferricyanide solution was added to the nanoparticles, the silver coating peeled off, and the silver ions were released. Consequently, PA signals were revealed from the internal AuNRs, providing a strong contrast in the PA images at an excitation wavelength of 750 nm. The released silver ions exhibited a strong bactericidal efficacy (>99.99%). In vivo monitoring of mice was used to verify the silver ion release by comparing the PA signals before and after silver etching (Figure 3c).

Low-dimensional nanomaterials with good biocompatibility and biodegradability have also been applied for contrast-enhanced PAI. De la Zerda et al., developed single-walled carbon nanotubes (CNTs) conjugated with cyclic Arg-Gly-Asp peptides, which showed high affinity to αvβ3 integrin in tumor angiogenesis [92]. They confirmed the PA signal enhancement at 690 nm excitation after injection of CNTs into U87MG xenograft tumors in mice. Cheng et al., demonstrated contrast-enhanced tumor imaging using tungsten disulfide (WS_2_) nanosheets [93]. They also verified the therapeutic effect through photothermal ablation in 4T1-bearing mice. Chen et al., achieved PA and FL dual-modal imaging using reduced nano-graphene oxide (rNGO) that can deliver ICG molecules with a high loading efficiency of π-π junction structure [94]. The same group also acquired contrast-enhanced PA images using molybdenum disulfide (MoS_2_) nanosheets [95]. They achieved a 54-fold signal enhancement at 675 nm laser illumination after injection of a single layer MoS_2_ into U87 xenograft tumors in mice.

### 3.2. Contrast Agents at High NIR-I (800–950 nm)

In the high NIR-I region, naphthalocyanine-based agents exhibit good optical absorption characteristics [96]. Zhang et al., demonstrated a family of nanoformulates consisting of naphthalocyanine dyes for contrast-enhanced PAI of the gastrointestinal tract of mice in vivo (Figure 4a) [97]. They synthesized multiple colors of agents with different absorption peaks using various types of naphthalocyanine dyes and obtained contrast-enhanced PA images of the lymphatic networks in rats (Figure 4b). For multispectral PAI, two different nanoformulated naphthalocyanines (absorption peaks at 707 and 860 nm) were separately injected into the left and right forepaws of mice, and dual-color PA images delineating the lymphatic path at each side were successfully acquired (Figure 4c) [98]. Recently, the same group demonstrated that nanoformulated naphthalocyanines could be used as theranostic agents for 4T1 breast cancer cells [99]. They increased the drug-to-surfactant ratio by removing free and unbound surfactants using a novel low-temperature process. The developed nanoformulations exhibited a strong optical absorption at 860 nm, providing contrast-enhanced PA images that could verify uptake by the tumor. The efficacy of the photothermal treatment was also evaluated based on light absorption and heat release. Choi et al., also developed naphthalocyanine-based nanodroplets for tumor resection using high-intensity focus ultrasound (HIFU) ablation [100]. They encapsulated naphthalocyanine dyes in perfluorohexane for effective HIFU ablation (Figure 4d). The resulting nanodroplets exhibited a maximum optical absorption at 850 nm (Figure 4e), and their in vivo imaging capability was validated after intravenous injection into MDA-MB 231 xenograft tumors in mice (Figure 4f). In addition, the therapeutic effect of HIFU ablation was verified by measuring the tumor size after treatment.

The strong optical absorption characteristics of AuNPs have also been applied to the development of PA contrast agents in the high NIR-I region. Bao et al., developed a PEGylated gold nanoprism (AuNPr) for detecting gastrointestinal cancer (Figure 5a) [101]. The anisotropic shape of AuNPr provided a strong absorption at 830 nm, making it suitable for contrast-enhanced PAI (Figure 5b). They obtained multispectral PA images of HT-29 tumors in mice before and after the injection of AuNPr. After injection, the PA amplitude in the tumor region increased (Figure 5c). Song et al., introduced AuNP-coated carbon nanotube rings (CNTRs) with enhanced light absorption and photothermal effects (Figure 5d) [102]. The high plasmonic coupling in the developed CNTR produced significantly increased light absorption at 808 nm (Figure 5e). After intravenous injection of AuNP-coated CNTR into U87MG xenograft tumors in mice, much stronger PA signals were achieved compared with those after the injection of pure CNTR (Figure 5f). In addition to contrast-enhanced PAI, the treatment efficiency was evaluated with photothermal ablation.

### 3.3. Contrast Agents at NIR-II (>1000 nm)

Several SPNs have been developed for extending the absorption wavelength to the NIR-II region. The energy levels and absorption wavelengths of these semiconducting polymers can be tuned by adjusting the combination of donor and acceptor molecules. For stable dispersion in biological media, hydrophobic SPNs have been synthesized using amphiphilic surfactants. Jiang et al., reported SPN for contrast-enhanced PAI in the NIR-II region [103]. The developed SPN showed broadband optical absorption between 700 and 1100 nm, which was achieved using a new polymer with a special structure comprising one donor and two acceptors (Figure 6a). The PA imaging capabilities of the NIR-I and NIR-II excitations (750 and 1064 nm, respectively) were evaluated on brains of rats after intravenous injection of SPNs. The signal-to-noise ratio (SNR) of the PA was 1.5-fold increased for the NIR-II excitation compared with the NIR-I excitation (Figure 6b). They also developed a metabolizable SPN that is readily degradable into phagocytes and decomposes into ultra-small (~1 nm) nanoparticles [104]. The developed SPN strongly absorbed light at ~1079 nm, providing contrast-enhanced PA images with an excitation wavelength of 1064 nm. Deep transcranial PA images were acquired, with excellent SNRs of 4.6 and 2.3 of tumors and vasculatures, respectively. Zhang et al., reported a similar SPN with strong absorption at ~1300 nm [105]. After injection of SPNs into 4T1 xenograft tumors in mice, contrast-enhanced PA images were acquired with a ~2-fold signal enhancement. The results demonstrated the effectiveness of two-acceptor SPN systems for tunable excitation wavelengths. Guo et al., synthesized an alternative two-acceptor SPN for high-resolution PAI [106]. The formulated SPNs exhibited strong absorption for 1064 nm excitation. Contrast-enhanced PA images of the whole cerebral cortex vasculature were acquired after intravenous injection of SPNs. The resulting images showed a high SNR (22.3 dB) at a depth of up to 1 mm through the intact skull.

Organic dyes have also been found to possess NIR-II-absorbing capability. Zhou et al., reported phosphorous phthalocyanine with an absorption peak at ~1000 nm (Figure 7a) [107]. The developed phthalocyanine-based dye could be used as a PA contrast agent, with an excitation wavelength of 1064 nm, which was generated using commercial Nd:YAG lasers (Figure 7b). Contrast-enhanced PA images were obtained from chicken breast tissue in an exceptionally deep position (~11.6 cm). Notably, phthalocyanine-based dyes were successfully visualized through a human limb (~5 cm) with an energy of 23 mJ/cm^2^, which is much lower than the MPE at 1064 nm (Figure 7c). The same group, Chitgupi et al., developed a surfactant-stripped micelle with a commercially available cyanine fluoroalkyl phosphate (CyFaP) salt dye for higher NIR-II absorption (Figure 7d) [108]. The effectiveness of the developed micelles was verified at a depth of 12 cm in chicken breast tissue, with an SNR of 24.3 dB (Figure 7e). Similar to the previous human experiment, tubes containing surfactant-stripped CyFaPs were imaged through the breasts of human volunteers (Figure 7f). The resulting images showed a strong imaging capacity at a depth of 2.6–5.1 cm of human breast tissue at an excitation wavelength of 1064 nm and energy of 21 mJ/cm^2^. Park et al., demonstrated a nickel dithiolene-based dye as an NIR-II-absorbing molecule at 1064 nm [109]. Hydrophobic nickel dithiolene-based dyes were dispersed in a biological buffer with an FDA-approved poly(lactide-co-glycolide) polymer via the nanoprecipitation method. The maximum detectable penetration depth for this PA agent was ~5.1 cm, with an SNR of 10.2 dB, in chicken breast tissue. The potential of this PA agent in preclinical and clinical investigations was demonstrated by contrast-enhanced PAI of sentinel lymph nodes, the gastrointestinal tract, and bladders in rats.

Inorganic nanoparticles have also been investigated as PA agents in the NIR-II region. These agents usually exhibit a better tunability of the bandgap for absorbing longer wavelengths of light. However, most of them have high aspect ratios, poor thermal stability, short blood circulation half-life, potential cytotoxicity, and systematic toxicity. Ku et al., reported the fabrication of copper sulfide nanoparticles with optical absorption at 990 nm as an inorganic NIR-II PA agent [110]. They demonstrated contrast-enhanced PA images of mouse brains using a 1064 nm Nd:YAG laser. Compared with copper, the high aspect ratio of AuNRs enables strong light absorption in the NIR-II range, as well as further tuning of the absorption spectra. Yim et al., demonstrated photostable NIR-II PA agents with a high aspect ratio (95 nm × 12 nm) in the form of AuNR-melanin hybrids (Figure 8a) [111]. The developed AuNR agents exhibited strong optical absorption over 1200 nm, which was used to generate PA signals with a laser excitation of 1064 nm (Figure 8b). Chen et al., reported a small, but high aspect ratio (50 nm × 8 nm) AuNR for NIR-II PAI [112]. This miniaturized AuNR could penetrate more easily into cancer cells and exhibited a high thermal stability compared with large AuNRs. They achieved contrast-enhanced PA images of tumors in mice with an approximately 4.5-fold signal improvement compared with large AuNRs. Zhou et al., reported another plasmonic AuNP that broadly absorbs light between 400 and 1300 nm [113]. The nanostructure was coated with methoxy PEG thiol and polydopamine to ensure structural integrity. Recently, an activatable NIR-II PA agent was developed using the polyoxometalate feature of molybdenum oxide (MoO_3_ (Mo^5+^/Mo^6+^)) (Figure 8c) [114]. The NIR-II absorbance of the MoO_3_ nanoparticles significantly increased when a portion of Mo^6+^ was reduced to Mo^5+^ (Figure 8d). Colon cancer tissues possess a high concentration of hydrogen sulfide (H_2_S), which reduces the oxidation state of MoO_3_ nanoparticles and eventually increases the PA signal of the tumor site. They demonstrated that endogenous H_2_S-activated NIR-II PA imaging could be employed on a tumor-xenografted mouse model (Figure 8e).

## 4. Conclusions

PAI is a promising biomedical imaging technique for visualizing the optical absorption characteristics of biological tissues in vivo. In addition to endogenous chromophores, such as hemoglobin, lipids, and melanin, exogenous contrast agents have been widely used for contrast-enhanced PAI. Moreover, exogenous agents have been modified to have various functionalities, including disease targeting and drug delivery capabilities, environment-related signal switching, and treatment ability. In vivo visualization of small animals using these contrast agents has been widely studied using various PAI systems [115,116,117]. According to the configuration of the PAI system, the penetration depth and spatial resolution of the resulting images are varied [118].

This review summarized contrast agents for PAI in terms of the absorption wavelength in the NIR range (Table 2). The wavelengths were divided into three ranges—low NIR-I (650–800 nm), high NIR-I (800–950 nm), and NIR-II (>1000 nm). Several organic dyes and nanostructures have been used in the low NIR-I region. They can visualize optically transparent tissues, but optical absorption by oxy- and deoxy-hemoglobin is dominant in this region; thus, the background signals from the surrounding tissues are relatively high. A number of organic dyes for low NIR-I PA imaging have been reported, including FDA-approved MB, BODIPY, and modified cyanine core structures. However, high NIR-I and NIR-II PA imaging agents have a smaller pool of organic dye structures; for example, a series of naphthalocyanine were reported to have high NIR-I absorption [97,98,99,100,101]. For the longer absorption spectrum in the NIR-II region, phthalocyanine, fluoroalkyl cyanine, and metal ion-chelated dithiolene dyes have been employed [107,108,109]. In the high NIR-I region, a higher laser fluence can be delivered to biological tissues because the MPE for the excitation laser increases at longer wavelengths. However, the two types of hemoglobin absorb light in this region, generating strong PA signals. Therefore, contrast agents that absorb NIR-II light have recently been developed. In addition to a high MPE, less photon scattering can significantly increase the imaging depth up to several centimeters. In comparison to visible light, NIR illumination is advantageous for PAI due to low background absorption and high MPE. Specifically, NIR-I excitation provides the lowest absorption from biological tissues, including water, protein, and lipids. PA imaging performance by NIR-I light can be maximized through the high water loading medium, such as the artery and vein systems. NIR-II light shows higher absorption than NIR-I in the tissues, mostly owing to water absorption, but it has the smallest scattering coefficient through many tissue types. PA imaging with NIR-II excitation is suitable for deep tissue imaging in the highly scattering media, such as the brain and bone.

In this review, we introduced various types of PA agents such as organic dye, SPN, low-dimensional nanomaterials, and inorganic nanoparticles. It will be noteworthy to address the pros and cons of each type of PA agent under specific conditions. Organic dyes usually have the smallest size, providing high biocompatibility in animal studies, which comes from rapid renal clearance. However, they sometimes suffer from irreversible oxidative photobleaching under the high-power laser, and it is usually difficult to modify the absorption spectrum of organic dyes over 1000 nm for NIR-II PA imaging agents. SPN also consists of the organic component only, so it has decent biocompatibility for in vivo monitoring. SPN is known to possess higher photostability than organic dye-based PA agents. The broad absorption spectra of SPNs would hinder the multiplexed PAI of multiple biotargets in biological systems. Low-dimensional nanomaterial and inorganic nanoparticles have relatively large absorption coefficients and endow higher PA sensitivity in complex media under modest illumination intensity. On the other hand, their poor biocompatibility and toxicity concerns will be the major bottleneck for in vivo PA application.

In addition to small animal studies, which are mainly discussed in this review, PAI can also be applied in human clinical research [119,120,121]. Recently, various clinical trials have been reported, particularly tumor-related studies. The majority of these studies used intrinsic chromophores to diagnose, classify, or monitor lesions. Oxy- and deoxyhemoglobins are typically used to measure hemoglobin oxygen saturation levels, which can indicate tumorous tissues [43,122,123,124]. However, these studies were limited to visualizing the tumor itself because light absorption is not dominant in tumors. Therefore, the use of contrast agents has been considered in clinical studies. In initial trials, FDA-approved ICG dyes were used for visualizing tumors [125], but the fast clearance of the dyes limits their use in quantitative analyses. Therefore, the contrast agents discussed in this review could be promising alternatives for visualizing, diagnosing, and classifying diseased tissues using features such as strong NIR absorption, disease targeting, disease-related activation, and therapeutic capabilities. For a successful clinical translation, further studies are necessary to evaluate the biocompatibility, biodegradability, toxicity, and solubility of PA agents and achieve FDA approval. In addition, to enhance the deep tissue imaging capability of PAI, advanced wavelength-tuning of nanoparticles with absorption peaks in the NIR-II region and the development of high-power lasers in the corresponding wavelength are required. Finally, the high photothermal conversion would widen the application area into the theragnostic field. With continuous efforts to develop contrast agents, PAI can be used as an essential tool for biomedical imaging in both preclinical small animal and clinical human studies.

## Figures and Tables

**Figure 1 biosensors-12-00594-f001:**
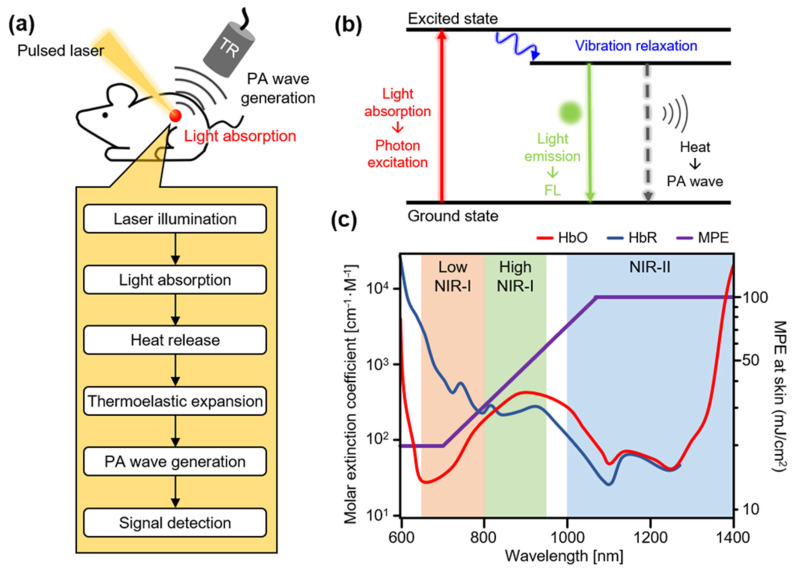
(**a**) Schematic illustration of principles of PA wave generation. (**b**) Schematic diagram of principles of signal generation through light absorption. (**c**) Molar extinction coefficient of hemoglobin and MPE according to wavelength in the NIR region. PA, photoacoustic; FL, fluorescence; TR, ultrasound transducer; NIR, near-infrared; HbO, oxy-hemoglobin; HbR, deoxy-hemoglobin; MPE, maximum permissible exposure.

**Figure 2 biosensors-12-00594-f002:**
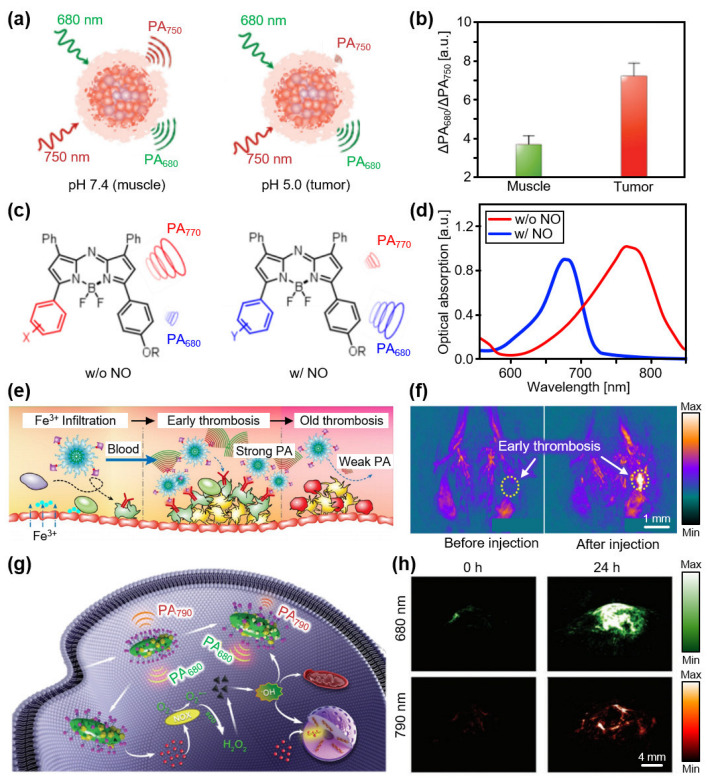
Contrast-enhanced PA imaging of organic nanostructures. (**a**) Schematic illustration of a pH-sensitive nanoprobe. (**b**) Ratiometric PA signals of a muscle (pH 7.4) and tumor (pH 5.0). Reprinted with permission from Ref. [84]. 2016, Wiley. (**c**) Schematic illustration of signal switching of an NO-activated nanoprobe. (**d**) Optical absorption spectra of a nanoprobe with and without NO. Reprinted with permission from Ref. [85]. 2018, ACS Publications. (**e**) Schematic illustration explaining the mechanism of PA signal generation in early thrombosis after injection of the PDI-based nanoparticles. (**f**) Contrast-enhanced PA images of mice before and after injection of PDI-based nanoparticles. Reprinted with permission from Ref. [86]. 2017, ACS Publications. (**g**) Schematic illustration of the PA signal switching that indicates ROS generation. (**h**) Contrast-enhanced PA images at 680 and 790 nm before and 24 h after injection. PA, photoacoustic; NO, nitric oxide; PDI, perylene diimide; ROS, reactive oxygen species. Reprinted with permission from Ref. [87]. 2018, Wiley.

**Figure 3 biosensors-12-00594-f003:**
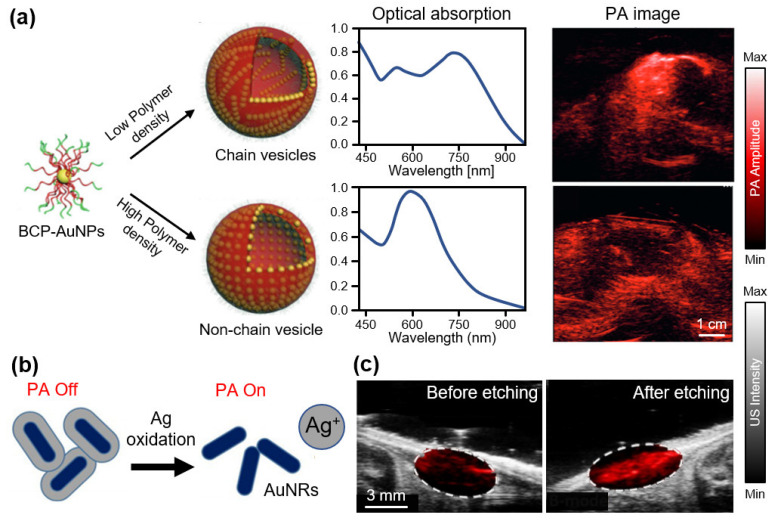
Contrast-enhanced PA imaging of AuNPs. (**a**) Schematic illustration of the synthesis of BCP-AuNP vesicles and their optical absorption spectra and contrast-enhanced PA images. Reprinted with permission from Ref. [90]. 2015, Wiley. (**b**) Schematic illustration of PA switching mechanism with and without Ag layer. (**c**) Overlaid PA and US images before and after Ag etching. PA, photoacoustic; US, ultrasound; BCP, block copolymer; AuNP, gold nanoparticle; AuNR, gold nanorod; Ag, silver. Reprinted with permission from Ref. [91]. 2018, ACS Publications.

**Figure 4 biosensors-12-00594-f004:**
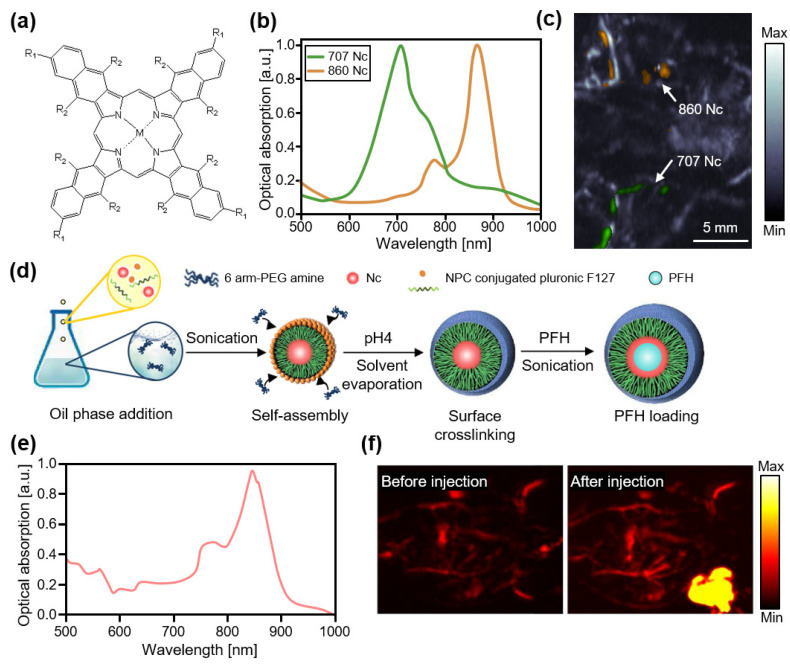
Contrast-enhanced PA images of Nc-based agents in the high NIR-I region. (**a**) Chemical structure of nanoformulated Nc dyes. 707 Nc is formulated with M = Zn, R_1_ = t-Bu, and R_2_ = H, while 860 Nc is formulated with M = 2H, R_1_ = H, and R_2_ = O-(CH_2_)_3_CH_3_. (**b**) Optical absorption spectra of two different nanoformulated Nc dyes. (**c**) Contrast-enhanced PA images of lymphatic networks in mice after injection of the nanoformulated Nc dyes into the right and left forepaws. Green and orange colors indicate PA signals from the nanoformulated Nc dyes, with a peak absorption at 707 and 860 nm, respectively. Reprinted with permission from Ref. [98]. 2015, Elsevier. (**d**) Schematic illustration for the synthesis of Nc encapsulated PFH. (**e**) Optical absorption spectrum of the encapsulated Nc dyes with PFH. (**f**) Contrast-enhanced PA images before and after injection of the encapsulated Nc dyes into mice. PA, photoacoustic; Nc, naphthalocyanine; NIR, near-infrared; PFH, perfluorohexane; NPC, 4-nitrophenyl chloroformate. Reprinted with permission from Ref. [100]. 2019, ACS Publications.

**Figure 5 biosensors-12-00594-f005:**
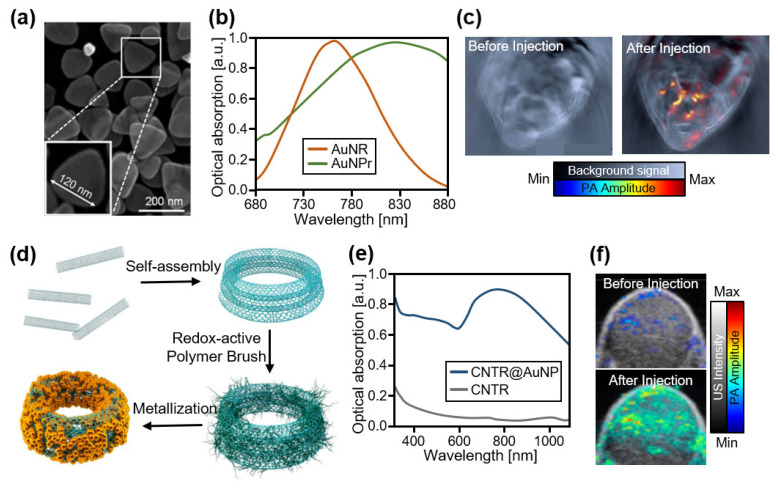
Contrast-enhanced PA images of AuNPs in the high NIR-I region. (**a**) The triangular structure of AuNPrs captured from an electron microscope. (**b**) Optical absorption spectra of AuNPrs and AuNPs, showing the shift. (**c**) Contrast-enhanced PA images of tumors in mice before and after injection of AuNPrs. Reprinted with permission from Ref. [101]. 2013, Wiley. (**d**) Schematic illustration for the synthesis of CNTRs. (**e**) Optical absorption spectra of CNTRs with and without AuNP coating. (**f**) Contrast-enhanced PA images of tumors in mice before and after injection of CNTRs. PA, photoacoustic; NIR, near-infrared; AuNP, gold nanoparticle; AuNPr, gold nanoprism; CNTR, carbon nanotube ring. Reprinted with permission from Ref. [102]. 2016, ACS Publications.

**Figure 6 biosensors-12-00594-f006:**
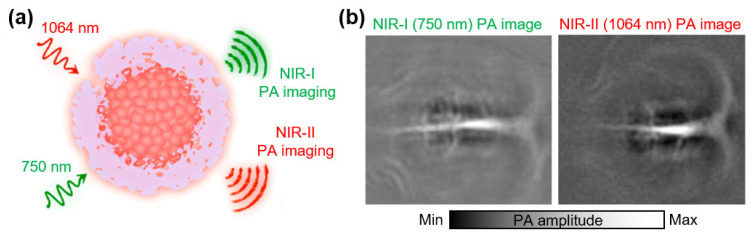
Contrast-enhanced PA images of SNPs in the NIR-II region. (**a**) Schematic illustration of SPNs for broadband PA imaging in NIR-I and NIR-II regions. (**b**) Contrast-enhanced PA images of a rat brain in vivo. PA, photoacoustic; US, ultrasound; SPN, semiconducting polymer nanoparticle; NIR, near-infrared. Reprinted with permission from Ref. [103]. 2017, ACS Publications.

**Figure 7 biosensors-12-00594-f007:**
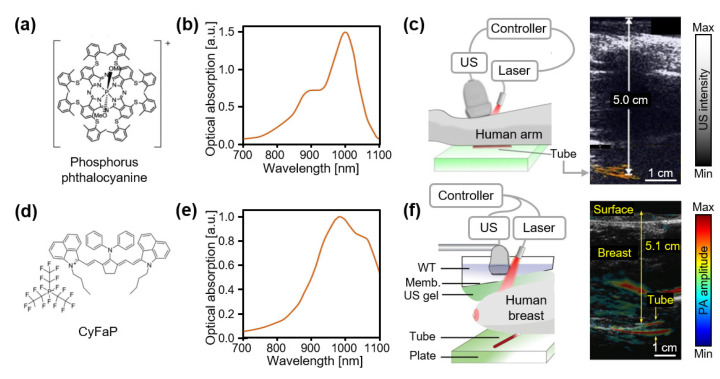
Contrast-enhanced PA images of organic dyes in the NIR-II region. (**a**) Chemical structure of a phosphorus phthalocyanine. (**b**) Optical absorption spectrum of a phthalocyanine-based dye. (**c**) Schematic illustration and contrast-enhanced PA images for evaluating deep tissue imaging through a human arm. Reprinted with permission from Ref. [107]. 2016, Ivyspring. (**d**) Chemical structure of CyFaP. (**e**) Optical absorption spectrum of surfactant-stripped micelle with a CyFaP salt dye. (**f**) Schematic illustration and contrast-enhanced PA images for evaluating deep-tissue imaging through a human breast. PA, photoacoustic; US, ultrasound; NIR, near-infrared; CyFaP, cyanine fluoroalkyl phosphate; WT, water tank; Memb, membrane. Reprinted with permission from Ref. [108]. 2019, Wiley.

**Figure 8 biosensors-12-00594-f008:**
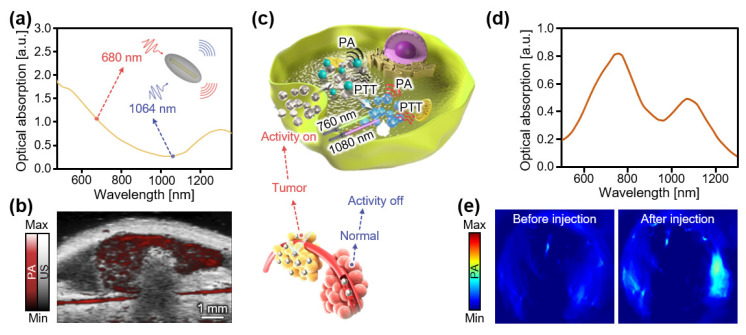
Contrast-enhanced PA images of inorganic nanoparticles in the NIR-II region. (**a**) Optical absorption spectra of AuNR-melanin hybrids. (**b**) Contrast-enhanced PA images of mice in vivo after the subcutaneous injection of nanoparticles. Reprinted with permission from Ref. [111]. 2021, ACS Publications. (**c**) Schematic illustration of PA activation of the MoO_3_ nanoparticles in the tumor. (**d**) Optical absorption spectrum of the MoO_3_ nanoparticles. (**e**) Contrast-enhanced PA images of tumors in mice before and after injection of the MoO_3_ nanoparticles. PA, photoacoustic; US, ultrasound; NIR, near-infrared; PTT, photothermal therapy; MoO_3_, molybdenum oxide. Reprinted with permission from Ref. [114]. 2021, ACS Publications.

**Table 1 biosensors-12-00594-t001:** Optical characteristics of organic dyes. ICG, indocyanine green; MB, methylene blue; EB, Evans blue; DMSO, dimethyl sulfoxide; $\lambda_{a}$, peak absorption wavelength; ϕ, fluorescence quantum yield.

	MB	EB	ICG
$\lambda_{a}$(m)	667	626	790
ϕ(%)	0.04	0.4	<1 in water 10 in DMSO
Ref.	[81]	[82]	[83]

**Table 2 biosensors-12-00594-t002:** Summary of PA contrast agents, including their optical absorption, PA signal generation, and applications. PA, photoacoustic; $\lambda_{A}$, peak optical absorption wavelength; $\lambda_{PA}$, excitation wavelength used for contrast-enhanced PA imaging; MB, methylene blue; EB, Evans blue; ICG, indocyanine green; SO, semiconducting oligomer; BODIPY, boron-dipyrromethene; SPN, semiconducting polymer nanoparticle; Au, gold; Ag, silver; AuNP, gold nanoparticle; AuNR, gold nanorod; AuNPr, gold nanoprism; SW-CNT, single-walled carbon nanotube; WS_2_, tungsten disulfide; rNGO, reduced nano-graphene oxide; Nc, naphthalocyanine; Pc, phthalocyanine; CyFaP, cyanine fluoroalkyl phosphate; NiPNP, nickel dithiolene-based polymeric nanoparticle, MoS_2_, molybdenum disulfide; MoO_3_, molybdenum oxide; SLN, sentinel lymph node; GI, gastrointestinal; LPS, lipopolysaccharide.

Range	Type	Base Material	$\lambda_{A}$ (nm)	$\lambda_{PA}$ (nm)	Main Application	Ref.
Low NIR-I	Organic	MB	677	635	SLN in rats	[76]
	Organic	EB	620	600	SLN in rats	[77]
	Organic	ICG	~700	668	SLN in rats	[79]
	Organic	SO with BODIPY dye	680	680	HeLa xenograft in mice	[84]
	Organic	aza-BODIPY dye	673	680	LPS-induced inflammation in mice	[85]
	Organic	PDI-based SPN	650	700	FeCl3-induced thrombus in mice	[86]
	Organic	PDI-based SPN	790	790	U87MG xenograft in mice	[87]
	Inorganic	AuNR	756	756	MDA-435S xenograft in mice	[89]
	Inorganic	Chain vesicle with AuNP	780	780	Subcutaneous layer in mice	[90]
	Inorganic	Au/Ag hybrid NP	750	800	Subcutaneous layer in mice	[91]
	Inorganic	SW-CNT	690	690	U87MG xenograft in mice	[92]
	Inorganic	WS_2_	~400	700	4T1 xenograft in mice	[93]
	Inorganic	rNGO	780	780	HeLa xenograft in mice	[94]
	Inorganic	MoS_2_	~500	675	U87 xenograft in mice	[95]
High NIR-I	Organic	Nc	863	860	GI tracts in mice	[97]
	Organic	Nc	860	860	SLN in rats	[98]
	Organic	Nc	860	860	4T1 xenograft in mice	[99]
	Organic	Nc	850	850	MDA-MB-231 xenograft in mice	[100]
	Inorganic	AuNPr	830	830	HT-29 xenograft in mice	[101]
	Inorganic	AuNP-coated CNTR	~800	808	U87MG xenograft in mice	[102]
NIR-II	Organic	SPN	1253	1064	Brain in rats	[103]
	Organic	SPN	1079	1064	4T1 xenograft and brain in mice	[104]
	Organic	SPN	1300	1280	4T1 xenograft in mice	[105]
	Organic	SPN	1160	1064	HepG2 xenografted ears in mice	[106]
	Organic	Pc	~1000	1064	Dye-containing tube through human arms	[107]
	Organic	CyFaP	1040	1064	Dye-containing tube through human breasts	[108]
	Organic	NiPNP	1064	1064	SLN, GI tracts, and bladder in rats	[109]
	Inorganic	Copper sulfide NP	990	1064	Brain and SLN in rats	[110]
	Inorganic	AuNR	~1280	1064	Subcutaneous layer in mice	[111]
	Inorganic	AuNR	~1050	1064	Prostate cancer xenograft in mice	[112]
	Inorganic	AuNP	~1300	1064	4T1 xenograft in mice	[113]
	Inorganic	MoO_3_	1080	1080	HCT116 xenograft in mice	[114]

## Data Availability

Not applicable.

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
