# Peer review of "Contrast Agents for Photoacoustic Imaging: A Review Focusing on the Wavelength Range"

_biosensors, 2022, doi:10.3390/bios12080594_

Round 1
Reviewer 1 Report
Han et al reviewed the contrast agents for photoacoustic (PA) imaging technique. Useful scientific information about PA imaging technique ranging from basic physics to application are described. Detail specification of the contrast agents are well listed and organized in terms of wavelength region and material system. The review would provide the deep insight into the selection of optimum imaging system for PA imaging applications, particularly providing the material constructs and wavelength selection. Based on this, we think that this review paper must be published in Biosensors. Few comments for further improvements are as below.
1. Specific advantages of the imaging with PA detection should be described comparing with the fluorescence (FL) detection. In page 2, authors described that PA can achieve both imaging depth (~10 – 20 mm) and the resolution (~100 – 500 μm), however, current fluorescence imaging technique such as wavelength-induced frequency filtering (Nature Nanotechnology 17, 643-652 (2022)) can even track the fluorescence signal for depths up to 5.5 ± 0.1 cm.
2. It would be fruitful if authors can describe the distinct application characteristics between each material types of contrast agent. Organic dye, organic nanoparticle, and inorganic nanoparticle should have different material stability, versatility, sensing performances, and functionalization availabilities, thus, researchers might need selection condition between the candidates for the PA imaging.
3. Not only for dye or nanoparticles, low-dimensional nanomaterials such as CNT or MoS2 have been also widely used as strong PA imaging agents with various micro/nano complex sensor construct (ex. Nature Nanotechnology 3, 557-562 (2008)). It also need to be described.
4. Key story of this review is focusing on the wavelength range. Thus, it would be great if authors can provide the comparison of contrast agents between each target nIR window in terms of application fields or imaging performances.
Author Response
Thank you for the valuable comments. Please find our response in the attached file.

Reviewer 2 Report
In this review, Kim and coworkers are reporting a concise summary of the Photoacoustic Imaging applications of the contrast agents specifically focusing on the wavelength range. The review seems to be beneficial for the Imaging fields are reasonably well-written. There are few grammatical and textual errors I noticed throughout the manuscript. Therefore, I advise authors to perform a careful revision on this aspect prior to the final publication. There are few comments I would like to add here.
(1). Can authors be specific about the penetration depth of the technique based on the appropriate references?
(2). Can authors provide a more detail summary of chemical structures of the PAI agents reported up to date?
(3). An appropriate diagram with the mechanism of the PA probes will be beneficial to the readers.
(4). For the section 3.1, is there any other additional PA agents that could be provided here?
(5). Can authors also explain how does the fluorescence quantum yield of the dye would affect the photoacoustic imaging efficiency?
(6). Authors must provide appropriate examples/structures for sections 3.2 and 3.3.
(7). In the conclusion section, authors can also suggest in brief future directions.
(7).
Author Response

(The authors gave the same response as above.)
